# A Socio-Inspired Methodology and Model for Advanced and Opportunistic Interactions between Industrial IoT Objects

Rim Kilani [1,*], Ahmed Zouinkhi [1], Eddy Bajic [2] and Mohamed Naceur Abdelkrim [1]

[1] MACS Laboratory LR 16ES22, National Engineering School of Gabes, University of Gabes, Omar ibn Elkhattab Street, Zrig Eddakhlania 6029, Tunisia; ahmed.zouinkhi@enig.rnu.tn (A.Z.); naceur.abdelkrim@enig.rnu.tn (M.N.A.)

[2] Research Centre for Automatic Control CRAN-CNRS 7039, Université de Lorraine, 54000 Nancy, France; eddy.bajic@univ-lorraine.fr

[*] Correspondence: rim.kiilani@gmail.com; Tel.: +21-6253-77180

**Abstract:** The concept of the Internet of Things (IoT) is widely discussed. IoT is one of the emerging technologies that have caught the attention of many researchers. The increase in the number of exchanges of services between heterogeneous or homogeneous connected objects with the integration of social networking concepts gives rise to the concept of the Social Internet of Things (SIoT). The SIoT concept contributes to the evolution of interactions between industrial objects by improving deterministic mechanisms towards intelligent interactions. The integration of the SIoT concept into the Industrial Internet of Things (IIoT) gives rise to the Social Internet of Industrial Things (SIoIT) and plays an important role in improving system performance in Industry 4.0. In this article, we propose an innovative methodology and a model of socio-inspired interaction between industrial communicating objects inspired by sociological approaches. Thanks to this model, socialized industrial communicating objects form a community of objects, autonomously and dynamically, by exchanging messages to know each other perfectly, and service requests between objects are executed adaptively according to the principles of social interaction governed by socio-inspired strategies and conditions. The model is implemented, tested and validated in a Netlogo multi-agent system simulation environment.

**Keywords:** Internet of Things; Social Internet of Industrial Things; socialized industrial communicating object; social community

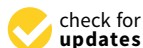



## 1. Introduction

The Internet of Things (IoT) is the environment that denotes the connectivity of physical objects. So describes a different universe of objects which are composed of elements of different natures such as sensors, actuators and smartphones, all of which have a unique identity [1]. These objects have distinct characteristics such as different hardware platforms, operating systems, associated standards, and different communication protocols. To meet the needs of their users, all devices must communicate with the other elements around it. A spatio-temporal-logical framework has been proposed in [2] which observes the relationships between IoT objects in physical space. He also observed that network structure is important for meaningful interaction and cooperation between different objects. The mobility of objects was another dimension that was considered. The fusion of the mechanisms of social networks and human behavior with the IoT opens a new vision of interactions between objects, the internet network and people where the exchange of information and relationships are governed by human rules and functionalities such as that of information exchange behavior [3]. Indeed, objects have an ability to establish a free social relationship. The integration of the concepts of principles of social networks and human interactions within the IoT gives rise to a flourishing field of research called SIoT (Social Internet of Things). SIoT is an active field of research in which social relationships

are established between various objects and users [4]. SIoT is about a "smarter and smaller" connected universe, spanning the smart home to the smart planet. Globally, interconnected social objects enable users, human and inanimate, to share information, services, and other resources [5]. In [6] the advantages of SIoT over IoT are further explained, even for the benefits of interconnection between social objects rather than smart objects. The social dimension of smart objects plays a very important role in improving network navigability and device accessibility, transforming network devices from smart objects into socially conscious smart objects. The architecture of SIoT guarantees the navigability of the network and the discovery of objects; services are carried out efficiently and as in human social networks the scalability is ensured [7]. The relationship between Device-to-Device (D2D) communications and SIoT has several advantages where social networks can increase the performance and efficiency of D2D communication. SIoT can help D2D communications to find relationships between social network users and to improve their skills when building them [8]. To advance machines, the SIoT concept could be integrated into the industrial Internet of Things to provide the formation of social relationships supporting human-machine communications and machine-machine interactions. The idea of integrating SIoT into industry is called the Social Internet of Industrial Things (SIoIT) [9]. With SIoIT, the individual machines that subscribe to the network share their working state with other machines; these smart machines can provide status updates to the social network. Consider a mining site mixed with a variety of capital-intensive equipment such as loaders, trucks, conveyors, and crushers made by different companies but interconnected through a social network. In the future world, especially in the future of manufacturing, to perform production tasks, intelligent machines will cooperate and communicate with each other via a social network very similar to human society [10].

The main contributions of our work are:

- The proposal of an innovative socio-inspired methodology and an associated model of interaction between industrial communicating objects. This model is dedicated to industrial communicating objects in the development of Industry 4.0.
- The transposition of the mechanisms and principles of social interactions from human beings to industrial communicating objects. With this transposition the response to services between objects is executed through principles of social interaction according to strategies and conditions.
- Socialized industrial communicating objects form a community in an autonomous and dynamic way by exchanging two types of messages to know each other perfectly.
- The implementation of our model on a multi-agent simulation platform to show its effectiveness.

In this article, after the introduction, we present the IIoT concept and system control. The following section presents Socio-inspired IoT interactions in which we show a synthesis of sociological approaches. From this synthesis we introduce a new model of principles of interaction between social industrial objects (SICO). Section 4 proposes the socio-inspired model of interaction between social communicating objects. Community management is introduced in the next section by explaining the notion of interpersonal distance as well as the creation, extension, reduction and change of communities. Section 6 introduces the exchange and execution of services between objects. The following section explains the implementation of our proposed model with the generalization of traffic messages for community building. The paper concludes with a conclusion in which we synthesize our work and identify future work.

## 2. Industrial Internet of Things and Control of Systems

A network is a collection of devices or systems connected to each other in which they can share resources with each other. Therefore, we can say that there are three different types of networks: centralized, decentralized, and mesh.

The centralized system is built around a central server that governs all requests from multiple nodes and assigns tasks to network nodes [11]. In this approach, the central

server controls full responsibility for the network. The central server performs all the processing operations and controls all the nodes connected to it; if the whole system will be unavailable, it is because the server breaks down. Therefore, if the single point fails, the whole system will fail. This problem is known as the single point of failure, and therefore the centralized system runs multiple gaps.

On the other hand, a decentralized system is a collection of interconnected nodes to form a single and integrated consistent network, and these nodes are autonomous. Without the need for a central server, nodes can communicate and connect with each other to provide various services and share information with other nodes in the decentralized system [12]. Unlike the centralized model, the decentralized system offers several advantages. There is no single point of failure, and the failure of one node will not affect the whole system. In addition, decentralized systems operate with a variety of different nodes. This distributed architecture is important in some applications such as the maritime sector [13] or the oil and gas sector [14].

According to the decentralized approach, the concept of Product-Driven System Control has provided a new model for the development of intelligent production systems, with extended capabilities in terms of responsiveness, adaptability, and production customization. Thanks to RFID (Radio Frequency Identification), RFID-tagged products take part in the decision making in the production process. A system controlled by the product changes the classic hierarchical and aggregated vision of the production planning and management function towards a more interoperable and intelligent system by defining the product as the controller of the resources of industrial companies [15]. This system presents a distributed perspective of decision making in which part of the decision is made locally and throughout the product life cycle. Thus, the necessary information is reduced and processed more locally.

The next step in decentralization in industrial systems control is afforded by the integration of Internet of Things concepts applied to industrial systems. The IIoT (Industrial Internet of Things) Driven Control approach allows the transition from centralized planning to extended decentralized control where each industrial IoT object and resource can contribute by interactions between objects and systems to the management and the monitoring and control of dynamic and complex systems [16]. Thanks to the exchange of data and services between objects and between objects and systems, benefits are important, for example in terms of process information availability for predictive maintenance in terms of local decision-making and reaction capabilities at the object/resource level within the process.

The increase in heterogeneous objects in IIoT, and the exchange of services in a deterministic Request/Response manner is insufficient, the socialization of objects makes it possible to improve intelligence by transposing the behaviour of human beings to industrial objects.

## 3. Socio-Inspired IoT Interactions

Social psychology is a field of scientific study that analyses how our social behaviour (thoughts, behaviours, and feelings) is influenced: by our personal psychological components, by various environmental stimuli, and by others. The research work in the fields of sociology and social anthropology on social behaviour between human beings formalize the forms of socialization and interactions between individuals according to the type of relationship.

### 3.1. Research Works in Sociological Interactions

Several works in the field of sociology and anthropology have been interested in models of interactions between individuals.

The works presented in [17] show that to achieve their own interests, everyone must choose a principle of interaction. They emphasize the principle of cooperation. Likewise,

anthropologists have developed this principle of interaction in the work presented in [18,19]. With this principle of cooperation, people act together to achieve common goals.

Tischler in [19] adds the principle of competition, which is a form of interaction in the modern world not only on the sports field, but on the market, in the education system, and in the political system.

In the resource exchange theory developed in [20], the authors have emphasized the principle of sharing, in which people share information and the principle of status or rank of authority, which is reflected in the relationship of prestige and respect. The principle of resistance has been developed in the work presented in [21,22]. According to this principle, the authors show where one does not make concessions to the other. Fiske et al., in [23], assume that people use four basic models to interpret, coordinate and think about most interactional aspects. They also introduce the concept of cost with the principle of market price. They assume that interactions between individuals are organized according to a common scale of values of the elements exchanged during the interaction.

### 3.2. A New Model of Principle of Interactions between Objects

After a survey study on the major sociological research works, we synthetized six standard sociological principles which make it possible to describe most relationships and social interactions between individuals. These principles are: Cooperate (Co), Dominate (Do), Compete (Com), Share (Sha), Resist (Re) and Monetize (Biz). Based on these interactions, we propose a transposition of these interactions towards communicating industrial objects.

Table 1 shows the objectives and behaviour of each principle of social interaction derived from the social theories stated above.

**Table 1.** Principles of people interaction with their transposition to industrial communicating objects.

| Principles | Social Life | Transposition to Communicating Objects |
|---|---|---|
| Cooperate (Co) | Cooperation, or the joint production of mutual benefits, is necessary when individuals cannot achieve specific results on their own. Individuals cooperate if each voluntarily acts in a way that contributes to the well-being of the other. Cooperation is called mutualism when it translates into net benefits for all parties involved. The individual becomes selfless in situations that benefit others. For selfish and rational (rationality) trading partners, their decision to cooperate is based on how the other party has behaved towards third parties in the past (memory). | Objects collaborate by sharing services for a common mission. There may be conditions for the performance of services during this collaboration. Each object has well-specified missions. Services are attached to each mission. A requested service belonging to the same mission is given unconditionally. Cooperation Strategies: Mutualism: cooperate by deriving a direct or indirect advantage. To characterize an advantage, the object derives a benefit from the use of the service by the applicant: for example, a product gives a temperature to a heating control system of the room where the product is located. Altruism: to cooperate without counting. Rationality: cooperate with a memory effect of previous cooperation, confidence between objects can be a decisive factor. |
| Dominate (Do) | Individuals are ordered according to a hierarchy, a status, which results in a ranking of authority, and accepted asymmetric relationships. Sociologists identify three forms of domination: Traditional domination (based on precedents) is based on the belief that tradition is sacred. Tradition gives the holder its legitimacy divine order: cannot be questioned. Personalized relationship: obedience is due to a person (respect) Lord/subject relationship Customary law). Charismatic domination (related to one person): based on the belief that an individual can be provided with exceptional qualities based on the sacredness and heroic character of a person. The group forms an emotional community of prophet-to-follower relationship that involves worship. It is not based on law, so it is unstable). Rational-legal domination: based on law, formal and written rules. A set of abstract rules that apply to particular cases. | Objects are assigned a rank of authority, which reflects their "charisma", "legitimacy", and "tradition". All objects are ordered in a hierarchical classification according to their rank of authority, thus imposing respected subordinate relationships in their respective interactions. The master/slave model is an ideal example of this mode of interaction between objects. Only a master can request a service from slaves that are forced to run. No master/master interaction accepted. The Client/Server model can satisfy this type of interaction when the rank of authority is played by client authentication. Domination strategy: By rank of authority |

**Table 1.** *Cont.*

| Principles | Social Life | Transposition to Communicating Objects |
|---|---|---|
| Compete (Com) | The act of competing is a special form of struggle. It is an indirect struggle between actors, the characteristics of which are as follows: (1) being aware of being in a struggle, (2) making parallel efforts to achieve a goal, (3) the result of which is exclusive: "if one wins, the other loses" (a criterion that differentiates it from emulation between actors). Rivalry between several people, several forces pursuing the same goal. | Objects competing for a service. Two modes: <br>• Several requesters-requesters/one executor-provider. <br>• A requestor-requester/several executors-providers. <br>Search for services competing with "At least" selection. |
| Share (Sha) | When an individual asks a loved one to help him perform a task that he does not know or cannot do on his own, the second is doing him a favour without asking for anything in return. In the capitalist world, it is an individual who asks another to help him with a task. The capitalist reflects and calculates what his service is "worth": either he exchanges it for another that he considers comparable (in money or in time, or in object), or he refuses, and their relationship ends there. If the services are not considered to be of "the same value", then one finds themselves in debt to the other. | The sharing of data, services and resources between all objects of a community is accepted de facto by all, either unconditionally or respecting the reciprocity of exchange of services. <br>Sharing Strategies: <br>1. No restriction (full open). <br>2. Conditioned by respect for equity and reciprocity of service sharing actions (equivalent to Fiske's Equality Matching). |
| Resist (Re) | A person resists when he refuses what is going on inside or outside, when he struggles with himself or others, when he denies what is. Oppose, slow down, counter, challenge, stand up, challenge, refuse, fight, deny. As you can see, resistance involves a movement to cancel, diminish the effect of another movement or of a force or an action. Resistance depends on the dispositional characteristics of individuals. For example, individuals who are highly authoritarian, optimistic, or have high self-esteem are usually the most difficult to persuade. | Strategies of resistance to service execution by an object: <br>• Delay the delivery of the service. <br>• Give partial information for a limited service. <br>• Give offended information. <br>• Denial of service. |
| Business (Biz) | Interactions between individuals are structured by economic costs, not necessarily monetary, and carried out according to a market mode of exchange based on a scale of values. Price is per service or barter, there are distinct ways to monetize data or services; directly, independently or by using a Data Broker as an intermediary, the exchange of information and improve performance. | The objects deliver their services against a predefined cost that is associated to each service according to a common rating system (NFT non fungible token, bitcoin, etc.). This introduces the notion of monetization of services execution between objects. Monetization of services can be decentralized and automated with high scalability, security, and confidence by application of blockchain smart contracts. Depending on the context, Bartering can be preferred, which involves the provision of one service by an object in return for another service from another object. |

### 3.3. From Deterministic towards Opportunistic Industrial Object Interactions

We propose a paradigm shift in the interactions between industrial objects by moving from deterministic relationships in an industrial environment constrained by networks, applications, to opportunistic relationships between more mobile entities (objects, resources, etc.), where each object will be able to have its own strategy of interaction which it will adapt to according to its environment and according to the nature and the strategies of the other objects which request it.

In the context of industrial piloting, the interactions summarized by a synthesis from an in-depth analysis of studies of social behaviour between individuals will be transposed to M2M (Machine-to-Machine) communications. In addition to socialization, M2M communication evolves from the classic interaction approach and tradition towards an innovative approach which is based on social paradigms inspired by social interactions.

- Deterministic classical approach. Without applying the predefined rules, the classic M2M interaction is defined with a service request/response as shown in Figure 1.

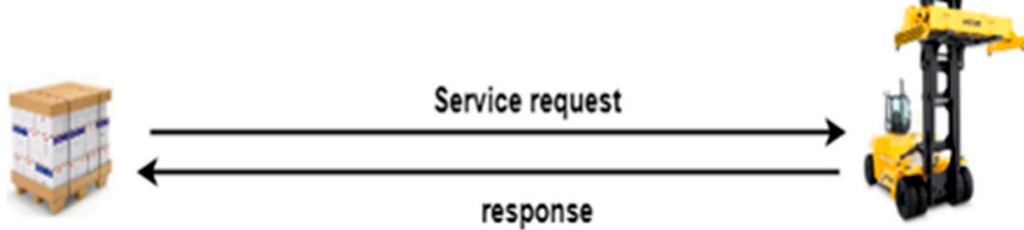

**Figure 1.** Deterministic Service/request interaction.

This classic and traditional type of exchange is not suitable, we therefore favor the emergence of the concept of autonomous intelligent objects. To interact with each other, these objects can simulate human behaviour. They will create a social framework by introducing the Social Internet of Things (SIoT) approach.

- Opportunistic interaction approach. This interaction is carried out by the application of the SIoT concept, i.e., the response to services is made through principles of interaction according to strategies and conditions as shown in Figure 2. From this approach, communication between M2M objects evolves from classic to innovative, therefore the transposition of social paradigms to objects around this smarter communication.

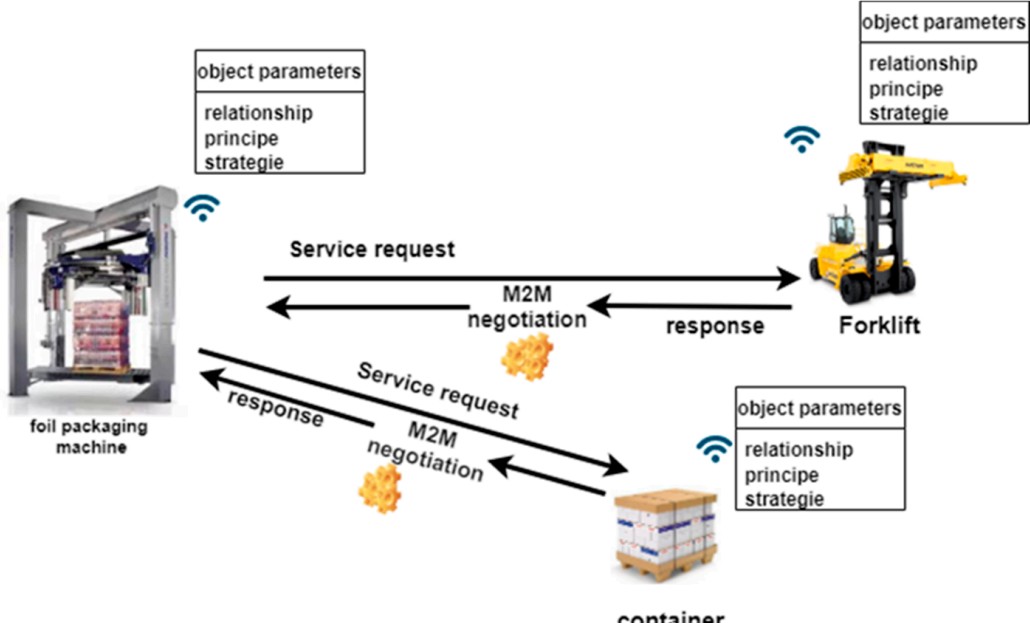

**Figure 2.** Interaction between objects by SIoT approach.

## 4. CoDoCOmShaRe.Biz: A Methodology of Socio-Inspired Interaction between Industrial Communicating Objects

We propose the codocomshare.biz model which is based on six principles of interactions between objects, as shown in Figure 3. All objects have their own characteristics or parameters that allow them to be individually differentiated from all other objects. These attributes come into play during a social interaction with another object. These parameters are used to determine the nature of the relationship between objects [24]. The presence of objects in a well-defined area forms a community. Each object that enters a community can request services from other objects in its community and these services can be performed according to one of these six principles.

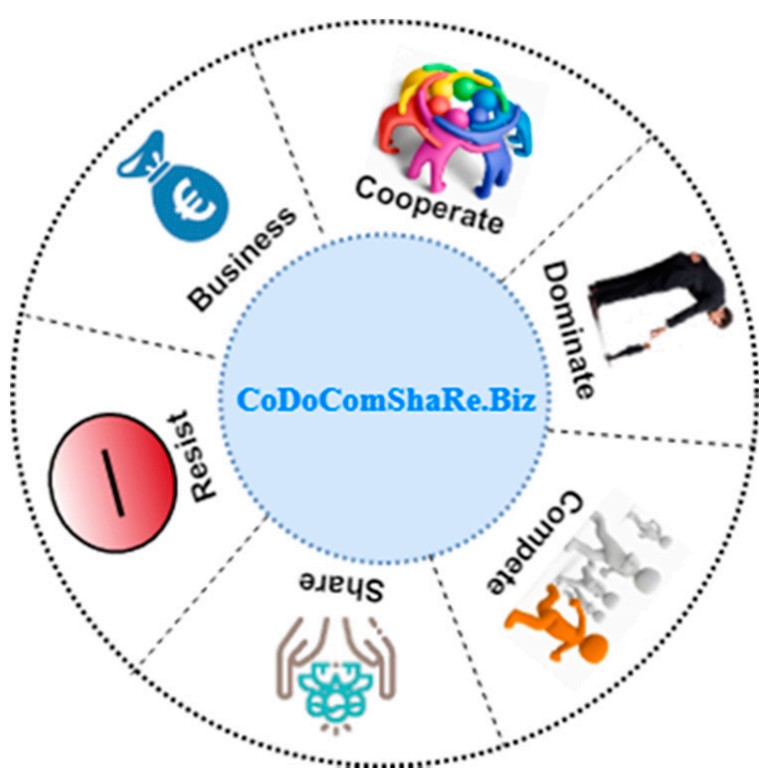

**Figure 3.** CoDoComShaRe.Biz principles of interactions.

### 4.1. Architecture of a Smart Socialized Industrial Communicating Object

Smart objects, which are self-contained objects of high sensing, processing, and network capabilities, are categorized as objects sensitive to activities, policies, and processes, and are seen as building blocks of IoT to meet functional needs [25]. By extension of the SICO model in the works of [26,27], a Social Industrial Communicating Object (SICO) can interact with other SICO in its environment by integrating relational paradigms inspired by human social interactions. In Figure 4, an architecture of the SICO is represented in which this architecture is based on the classical architecture. A SICO is characterized by elements inspired by the social approach, while the social aspect is defined by three main points, which are: the attributes of an object, services, and social group.

According to Atzori [28] the recording of friend information of an object like identifier (ID), metadata about its use and type (e.g., logistics, sport, sensor, etc.), the types of relations (POR, CWOR, SOR, CLOR, OOR) are made in a table called the Friend Table (FT). The latter is kept by the object and recorded in its virtual entity. This entity can be implemented in home repositories in the cloud or at the edge of the network infrastructure. Updates to their data structures are performed with caused data from the corresponding physical entity.

In our work, each intelligent communicating object keeps a table called a "community table" which presents the characteristics (ID, AID, OID and FC) and the relationship (POR, CWOR, SOR, CLOR, OOR) established with the other objects that are in its interpersonal distance. The object can remember objects it has seen before. The objects are not stable, therefore the number of objects which have been found in the interpersonal distance varies, and this gives the table dynamicity.

Figure 4 presents the functional model of the social industrial communicating object. In this model we find the classic aspect represented in the color green and the social aspect represented in blue which presents the services related to the principles of CoDoComShaRe.Biz interaction and the established type relationships with other objects. The SICO object can communicate with other SICOs, Operators, or local or remote Base Stations.

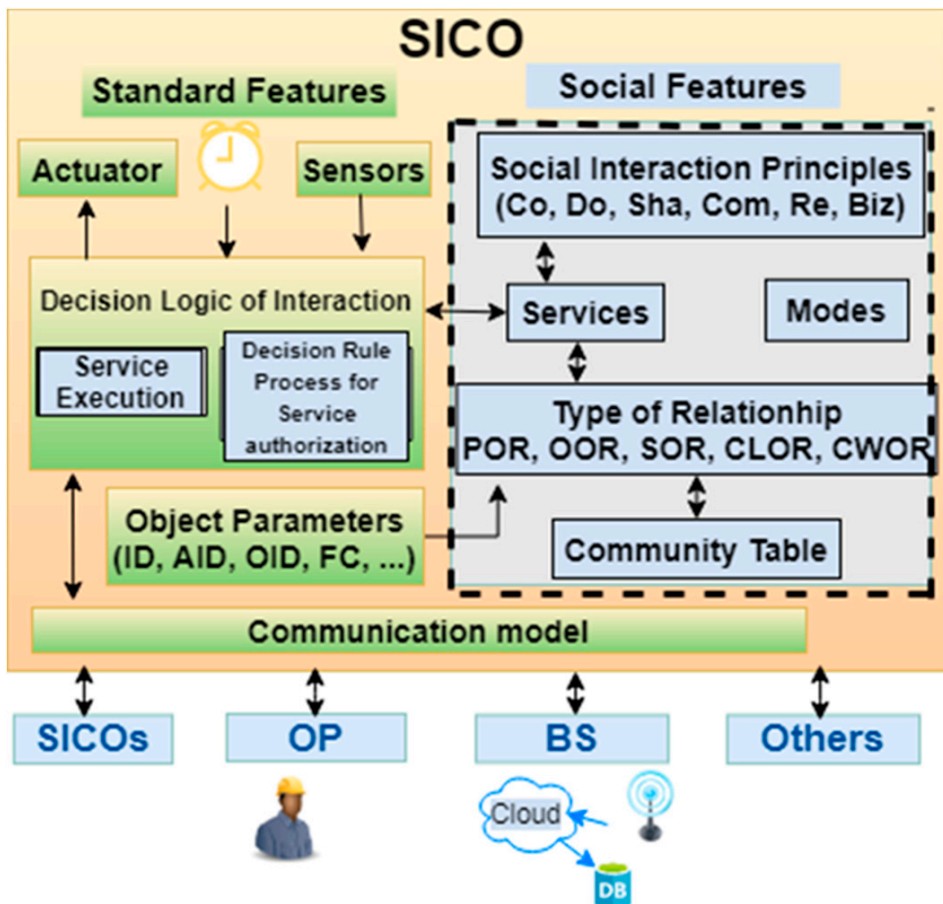

**Figure 4.** Functional model of a community table.

*4.2. Typology of Object Relationships*

Social relationships are evolved so they are not limited to human interaction, and they can be extended to machine-to-machine (M2M) interactions.

Therefore, the improvement in service research and resource discovery is due to the integration of social relationships between objects [6].

In the presence of social interaction, all individuals of the same community can make a contact. Contact between them establishes a type of relationship that can influence the nature of the interaction because everyone has their own characteristics. By being inspired by human social relationship. Refs. [29,30], presented five types of relationships that can be established between IoT objects. The logic for establishing these relationships is shown in Figure 5.

Parental Object Relationship (POR): In this relationship, smart objects are homogeneous, belonging to the same family, manufactured by the same manufacturer, belonging to the same production batch and in the same period of time.

Ownership Object Relationship (OOR): Finds only heterogeneous objects belonging to the same owner.

Co-Work Object Relationship (CWOR): to accomplish a task or a common work, the objects collaborate and the CWOR relationship established between them.

Social Object Relationship (SOR): The social relationship is characterized by the physical meeting between the owners of connected objects where the connected objects come into contact intermittently or continuously.

Co-Location Object Relationship (CLOR): the objects that are in the same place or in the same environment can create this relation.

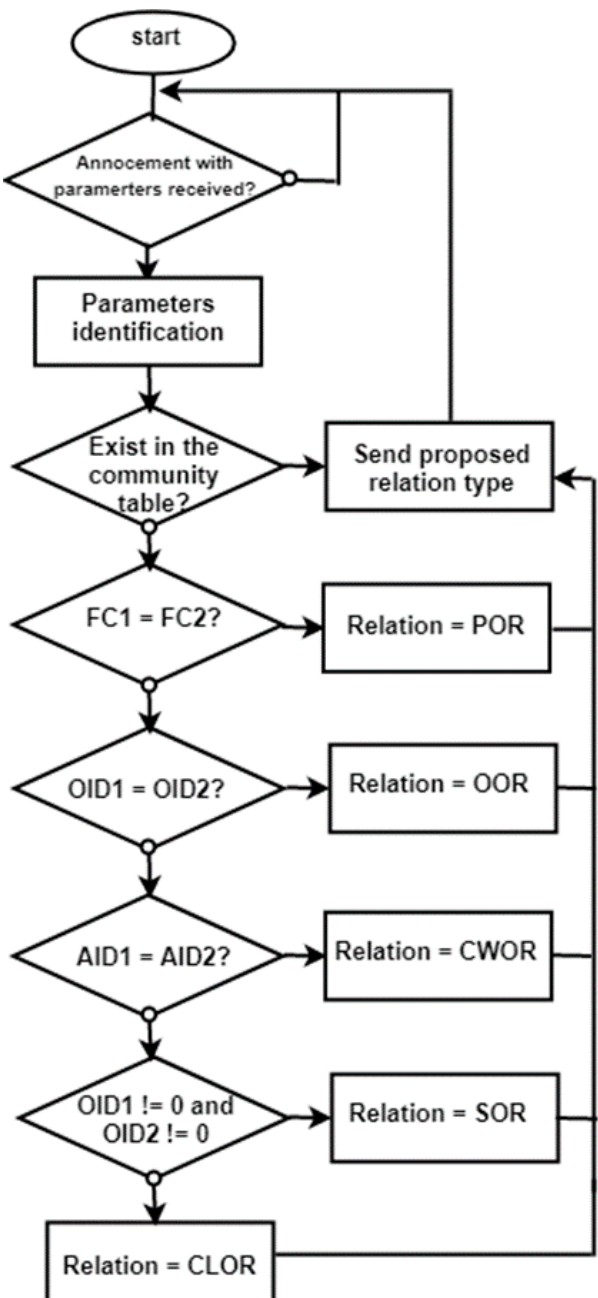

**Figure 5.** Type of relationship between communicating objects.

*4.3. Interaction Sequence*

Figure 6 shows an interaction sequence, which is a set of reciprocal actions between two smart objects based on message exchanges and providing services to resume on demand from one of the two smart objects.

At first, an object announces itself in a Push mode, which is well suited to the autonomy and mobility of objects [31]. This mode is the appropriate message communication protocol for IoT devices because it is built into a low bandwidth network [32,33]. Once it detects an object nearby, it asks for its settings to determine what kind of relationship it can make. Each object then discovers the list of services offered.

The interaction sequence contains two main steps: the creation of community and Request/Execution of service. The first is done by community messages and service messages and the second by service request or service execution.

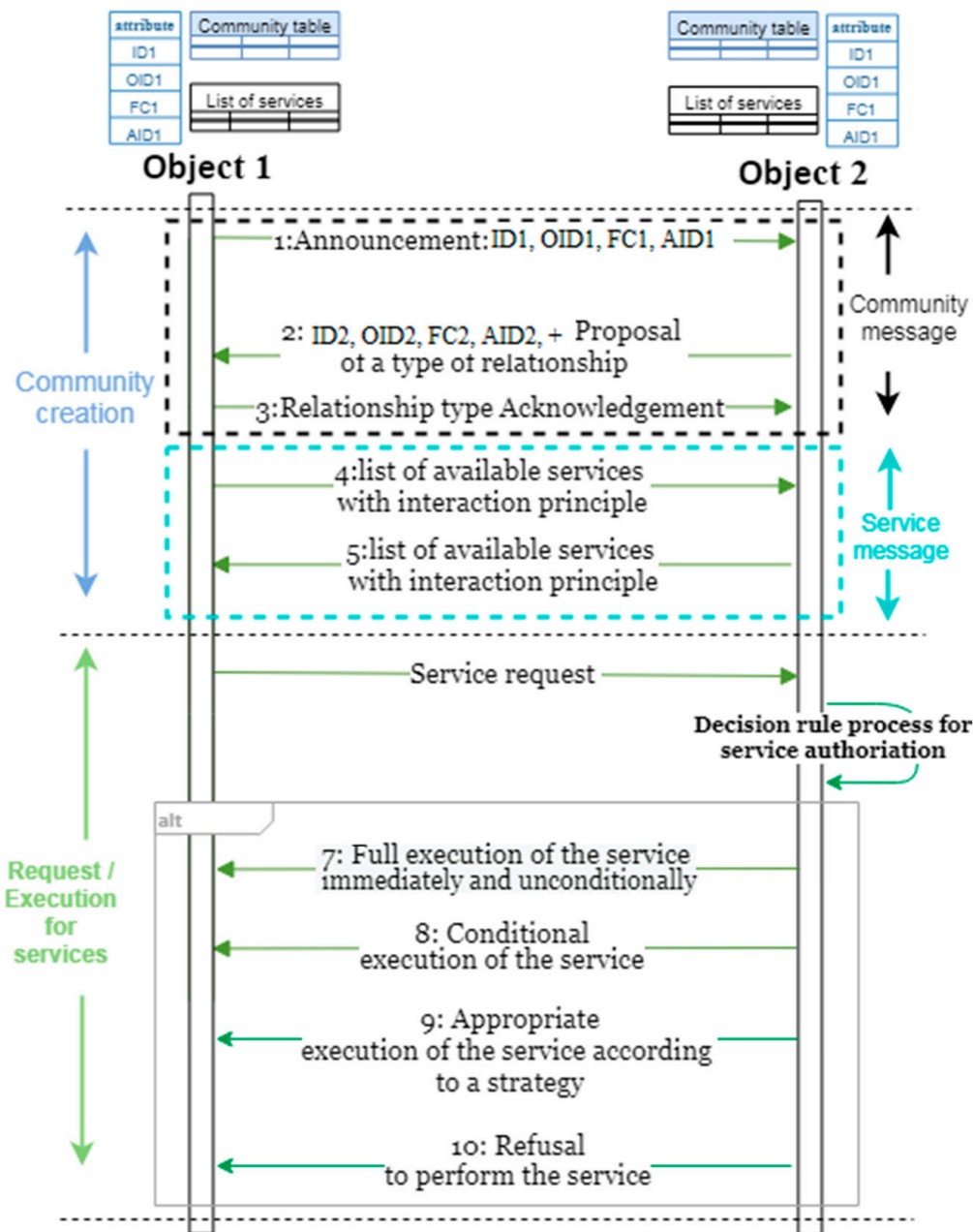

**Figure 6.** Interaction sequence.

## 5. Management of a Community of Communicating Objects

Each object that communicates with other communicating objects according to rules defined with a distance is called interpersonal distance, and this is by analogy to human beings. Each object has a list of services that can run them. For the discovery of services in the decentralized approach, from the same community each object requests and knows the services of all the other communicating objects. The notion of distance is important in communication, so several types of distance can be distinguished as communication distance and Interpersonal distance. The communication distance is the distance in which a communicating object has a free communication coverage area. In this zone the transmission power of an object is very important. The two notions of communication distance and interpersonal distance are integrated into the concept of interpersonal distance, which has a social impact on the relationship between individuals.

### 5.1. The "Interpersonal Distance" Concept

The notion of interpersonal distance is the distance in which each contact is characterized by a distance that must be respected for the contact to be effective and comfortable for all. Some people need the space around them to feel safe, and others appreciate closeness and even physical contact. By adapting to this distance, to the lack of passion or mistrust or on the contrary we ensure the desire to engage in conversation, the desire for intimacy. Things like people, relationship type, and some personal factors influence variation in interpersonal distance. As shown in Table 2, Edward T. Hall experimentally determined the existence of four physical distance zones in humans: they are the intimate, personal, social, and public areas. Each of the distances has two modes, near and far [34].

**Table 2.** Physical distance zones.

| Distance Zones | Near Mode | Distant Mode |
| --- | --- | --- |
| Intimate area | Body to body | 15–40 cm |
| Personal area | 45–74 cm | 75–125 cm |
| Social area | 1.25 m–2.10 m | 2.10–3.60 m |
| Public area | 3.60 m–7.50 m | 7.50 m and beyond |

The established distance zones are represented in Figure 7. For example, for a COVID-19 safe area, one has to keep a distance of at least two meters from others. This example can be considered as a social area.

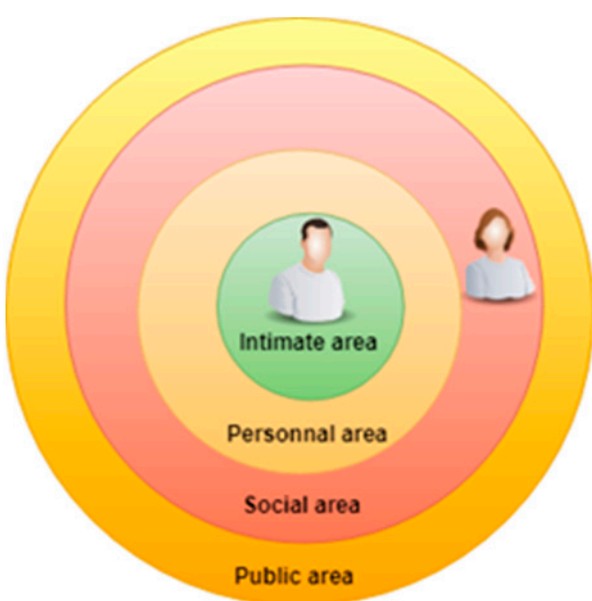

**Figure 7.** The different interpersonal areas.

Any meeting is characterized by a distance that it is preferable to respect so that the contact is effective and comfortable for everyone. Some people value closeness and even physical contact, and others need the space around them to feel safe. By adjusting to this distance, we show a willingness to engage in conversation, a desire for privacy, or on the contrary, a lack of interest or mistrust.

In our work, the notion of distance is important in communication. A communicating object has a free communication coverage area, mainly conditioned by its transmission power: this is its Communication Distance.

In Figure 8, 3 objects are surrounding Object 1.

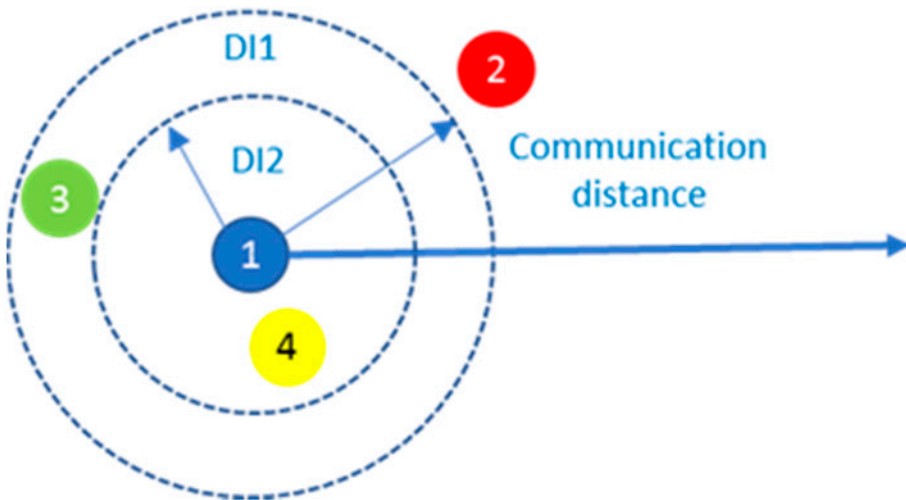

**Figure 8.** Communication distance.

We notice that object 2 does not belong to the interpersonal distance of object 1 so it receives messages from object 1, but they cannot communicate with each other. Objects 3 and 4 can communicate with object 1, with a possible treatment differentiation according to the interpersonal distance DI1 and the interpersonal distance DI2 from what they are in relation to the object 1.

On the other hand, messages from other objects will not be processed unless they are at a defined distance called Interpersonal Distance. An object can have several decreasing Interpersonal Distances, which can influence the social behaviour of the object on its relations of service towards the other objects present in this area.

Figure 9 defines the interpersonal distance in which object 2 receives messages from object 1, but object 1 does not respond to messages from object 2, unlike object 3 which is in the interpersonal area, that is, object 1 can communicate with each other.

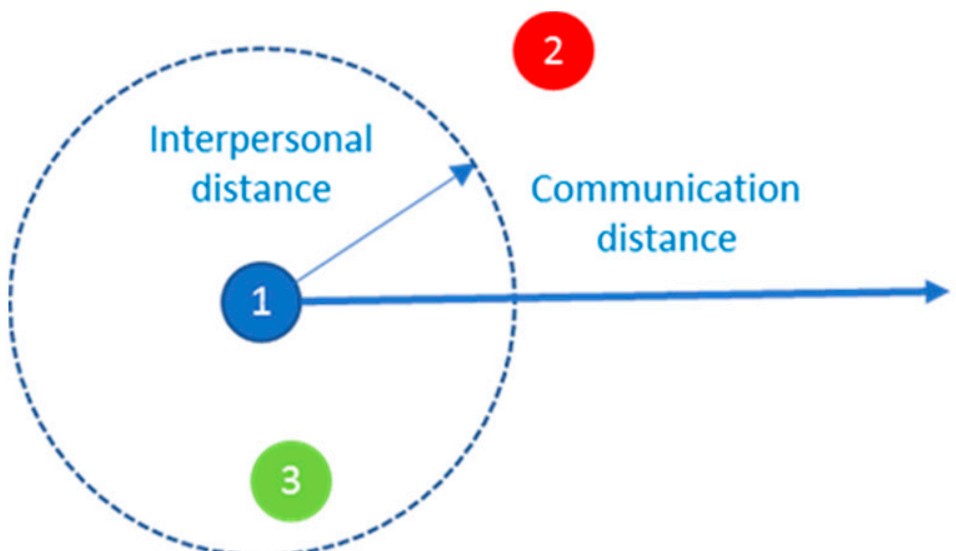

**Figure 9.** Interpersonal distance.

Each communicating object defines and manages its relationship with the objects of its community; the other objects are included in a well-defined distance which is known as the interpersonal distance. Depending on the relationship established between them, an object can invoke services from the other object, or perform its own services at the request of the other object. For this, each object must know the list of services of the other objects.

Figure 10 illustrates two objects forming a unique community because each object is in the interpersonal distance of the other.

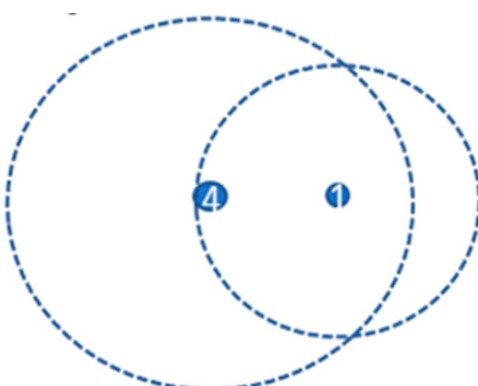

**Figure 10.** Objects in a unique community.

On the contrary, the objects represented by Figure 11 do not form a community.

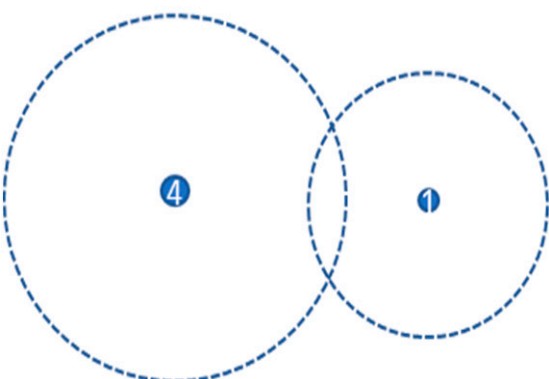

**Figure 11.** Communicating object in community.

Each Object has two tables to manage its relationships in the community of objects: a community table and a list of services as shown in Figure 12.

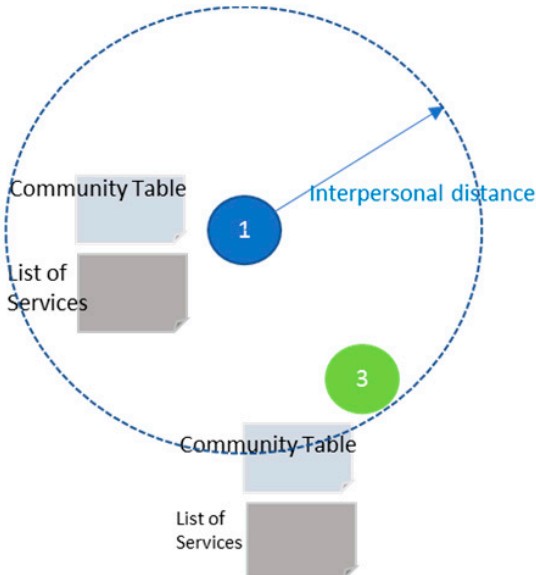

**Figure 12.** Communicating object.

**Community Table:** A communicating Object memorizes the characteristics (AID, OID, FC) and the relation established with the other Objects which are in its Interpersonal Distance. Thus, it can remember Objects it has encountered before.

**List of Services:** A communicating Object memorizes the lists of the services of the other Objects, which are in its Interpersonal Distance, according to a distributed approach. Unlike the centralized approaches, it does not remember the services of the Objects that it has crossed before. This assures the management of dynamicity of change of services the objects may have during their life.

*5.2. Creation of Communities of Objects*

An Object creates its own community with the objects that are situated within its interpersonal distance.

The overlapping of interpersonal distance is not sufficient to make object entering in a community of the other object.

The objects presented in Figure 13 form specific communities of Objects in that:

- Each object has its own community table.
- Each object has a specific list of services available on all other objects in its community.

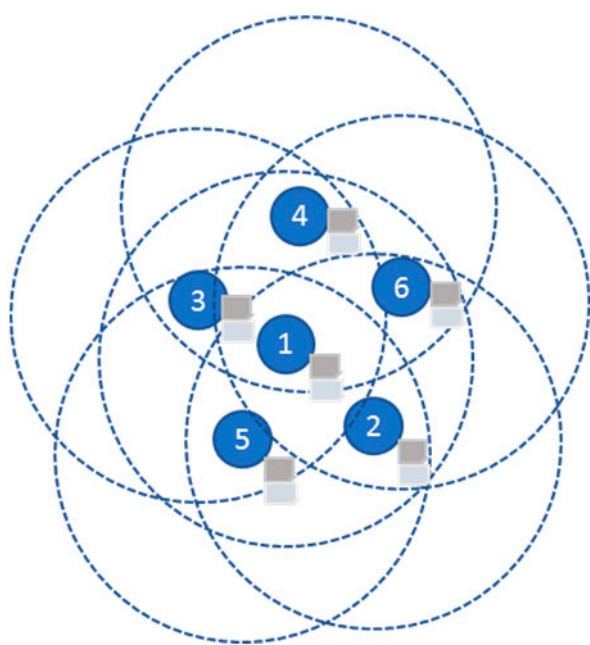

**Figure 13.** Multiple object communities.

Based on the schema of Figure 13, we can deduce the composition of the community of each of the 6 objects (Table 3), which are all different.

**Table 3.** Each Object has its own community.

| Object | Object Community |
| --- | --- |
| 1 | 2, 3, 4, 5, 6 |
| 2 | 1, 5, 6 |
| 3 | 1, 4, 5 |
| 4 | 1, 3, 6 |
| 5 | 1, 2, 3 |
| 6 | 1, 2, 4 |

It can be considered the case when the objects do not have the same interpersonal distance due to a limitation of their communication range, the restriction of the working area of objects, depending on the application case. So, this will also affect the community area of objects.

The case where all objects belong to the same community is achievable when all objects are situated in the interpersonal distance of all the other objects as is presented in Figure 14.

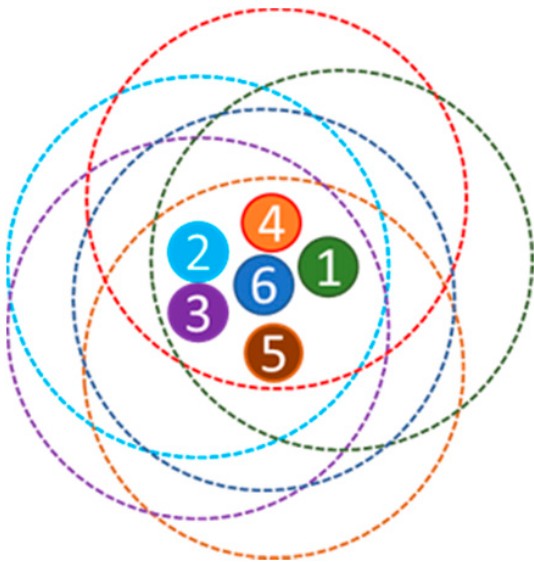

**Figure 14.** A unique community of Objects.

### 5.3. Extension of a Community of Objects

Each Object has its own community, which may be different from the community of other objects that are in its community or that are close to it. There are communities of objects in the same physical space. The community of an object changes each time an object "meets" another object (Figure 15). In order to ensure that that there is only one community between objects within the same space, an Object does not enter a community, but it will have in common the objects of the communities of other objects. There is an "overlap" between individual communities of the objects of a community. Table 4 shows the community formed by each object before the arrival of Object 6 as well as after arrival.

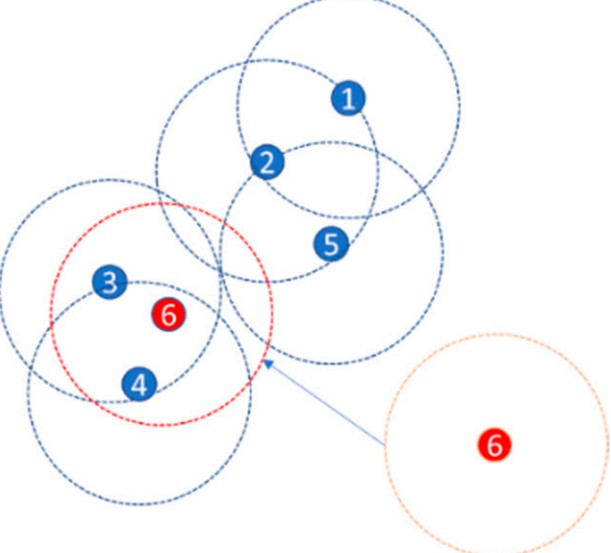

**Figure 15.** Extending a community.

**Table 4.** Multiple communities of objects.

| Object | Community before Arrival | Community after Arrival |
|:---:|:---:|:---:|
| 1 | 2 | 2 |
| 2 | 1, 5 | 1, 5 |
| 3 | 4 | 4, 6 |
| 4 | 3 | 3, 6 |
| 5 | 2 | 2 |
| 6 | Nil | 3, 4 |

*5.4. Reduction of a Community*

The conditions and events that lead to the reduction of an object community, i.e., the reduction of the number of objects in its community table are: (a) an object is deactivated (OFF), (b) the battery is exhausted, (c) the object moves and leaves the zone of interpersonal overlapping of other objects, or, finally (d) the object deliberately enters "transparent" mode.

In Figure 16 the object 6 is moving and thus leaving the community of object 4 and object 5 because it leaves their interpersonal zones. Therefore, object 6 gets "out of reach". Objects 4 and 5 realize that object 6 has disappeared because they no longer receive its announcement message which can be considered as equivalent to the "I am alive" message in the TCP protocol.

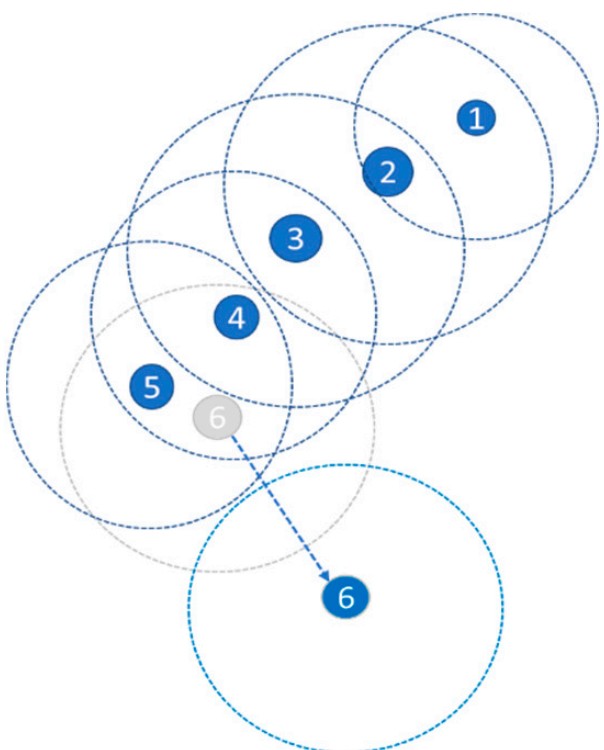

**Figure 16.** Reduction of a community.

The object makes it possible to regularly inform (configurable frequency) the other objects of its presence. If an object does not receive, within a configurable limited time (lifetime, lease, etc.), a message announcing an object from its community, then this is removed from the community of the object and maintained in the community with an "absent" status and the timestamp of the last announcement.

Table 5 shows the communities of objects before and after the separation of object 6.

**Table 5.** Community before and after the departure of object 6.

| Object | Community before Departure | Community after Departure |
|--------|----------------------------|---------------------------|
| 1 | 2 | 2 |
| 2 | 1, 3 | 1, 3 |
| 3 | 2, 4 | 2, 4 |
| 4 | 3, 5, 6 | 3, 5 |
| 5 | 4, 6 | 4 |
| 6 | 4, 5 | Nil |

*5.5. Change of Community*

An object moves, it is nomadic and can change communities, for example, object 6 in Figure 17 leaves its community to enter another community.

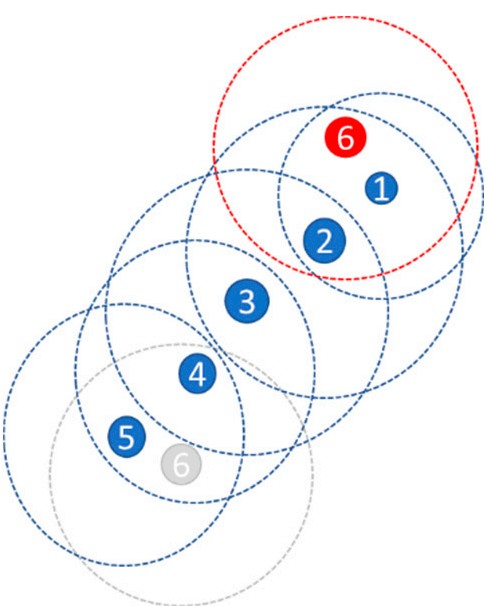

**Figure 17.** Change of community.

Object 6 is mobile and leaves the overlap area of Object 4 and 5 to enter the overlap area of objects 1 and 2. Object 6 keeps the memory of its old community and updates its new one community with Objects 1 and 2. Objects 1 and 2 update their community with Object 6.

Table 6 illustrates the communities before and after the change of community of object 6.

**Table 6.** Community before and after the change of community of object 6.

| Object | Community before the Change | Community after the Change |
|--------|------------------------------|-----------------------------|
| 1 | 2 | 2, 6 |
| 2 | 1, 3 | 1, 3, 6 |
| 3 | 2, 4 | 2, 4 |
| 4 | 3, 5, 6 | 5, 6, 3 |
| 5 | 4, 6 | 4, 6 |
| 6 | 4, 5 | 4, 5, 1, 2 |

## 6. Services Exchange

According to [26], an object does not choose the interaction principle. Each service offered by an object is linked to one or more FISKE principles and types of relationships, which are defined in the configuration of the object.

In our work, each service offered by an object is linked to one or more CoDoComShaRe.Biz principles which are defined in the configuration of the object. Depending on the type of relationship (which is determined with a deterministic algorithm) with another object discovered by announcement, a strategy of interaction principle which is linked to the requested service is executed. The requested service will be executed according to the principle attached to it and the relationship established with the service requesting object as indicated in the influence table (Table 7).

**Table 7.** Influence matrix of relationships and principles on the activation of requested services.

| Service | | Principal Interaction | | | | | |
|---|---|---|---|---|---|---|---|
| | | Cooperate (Co) | Dominate (Do) | Compite (Com) | Share (Sha) | Resist (Re) | Bizness (Biz) |
| Social relation | Co-location CLOR | XCo = Sa | XD | XCom = So | Xsh | XR = 0 | XMo = P |
| | Social SOR | XCo = Sa | XD | XCom = So | Xsh | XR = 0 | XMo = P |
| | Co-work CWOR | XCo = Sc | XD | XCom = So | Xsh | XR = So | XMo = P |
| | Ownership OOR | 1 | XD | XCom = 0 | 1 | XR = Y | XMo = P |
| | Parental POR | 1 | XD | XCom = 0 | 1 | XR = Sreq(t + dt) | XMo = P |

■: Full execution of the service immediately and unconditionally; ■: Conditional execution of the service; ■: Appropriate execution of the service according to a strategy; ■: Refusal to perform the service.

The parameters Xco, XD, XCom, XSh, XR and XMo represent the service execution result when the interaction principle is Cooperate, Dominate, Compute, Share, Resist and Business, respectively, and will be determined according to the following equations. The parameters are determined by the following equations:

$$X_{Co} = \begin{cases} 1 \; if \, (R = OOR) OR (R = POR) OR (Av = 1 \; AND (R = CLOR) or (R = SOR)) \\ 0 \; otherwise \end{cases} \tag{1}$$

With:
*Av*: is a benefit indicator for an object (0 or 1).
*R*: Social Relationship.

$$X_D = \begin{cases} 1 \; if \; \Delta R > 0 \\ 0 \; if \; \Delta R \leq 0 \end{cases} \tag{2}$$

With:

$$\Delta R = Rank_{requester} - Rank_{provider}$$

$$X_{Com} = \begin{cases} S_o \; si \; (R = CWOR) OR (R = SOR) OR (R = CLOR) \\ 0 \; if \, (R = OOR) OR (R = POR) \end{cases} \tag{3}$$

With:
$S_o$: Obfuscated service strategy.

$$X_{Sh} = \begin{cases} 1 \; si \; ((R = OOR) OR (R = POR)) OR ((\Delta c > 0) AND ((R = CWOR) OR (R = SOR))) \\ 0 \; otherwise \end{cases} \tag{4}$$

With:

$\Delta c$: Exchange Strategy with Equality Matching.

$$X_R = \begin{cases} 0 \ if \ (R \ = \ CLOR \ or \ R \ = \ SOR) \\ S_p \ if \ R \ = \ OOR \\ S_o \ if \ R \ = \ CWOR \\ S_{Req}(t + \Delta t) if \ R \ = \ POR \end{cases} \tag{5}$$

With:

$S_{Req} \ (t + \Delta t)$: delayed Strategy with $\Delta t$.
$S_p$: Partial strategy.

$$X_M = \begin{cases} 1 \ if \ payment \ is \ done \\ 0 \ if \ payment \ is \ not \ done \end{cases} \tag{6}$$

### 6.1. Sequence Execution of Service

The execution sequence of a service is carried out according to the flowchart shown in Figure 18. A service to be performed must go through the "dDecision rule process for service authorization" step. This step is used to assign a service the result "Service Authorized" or "Service Not Authorized". If the result is "Service Authorized", execution proceeds to the "Service execution" step.

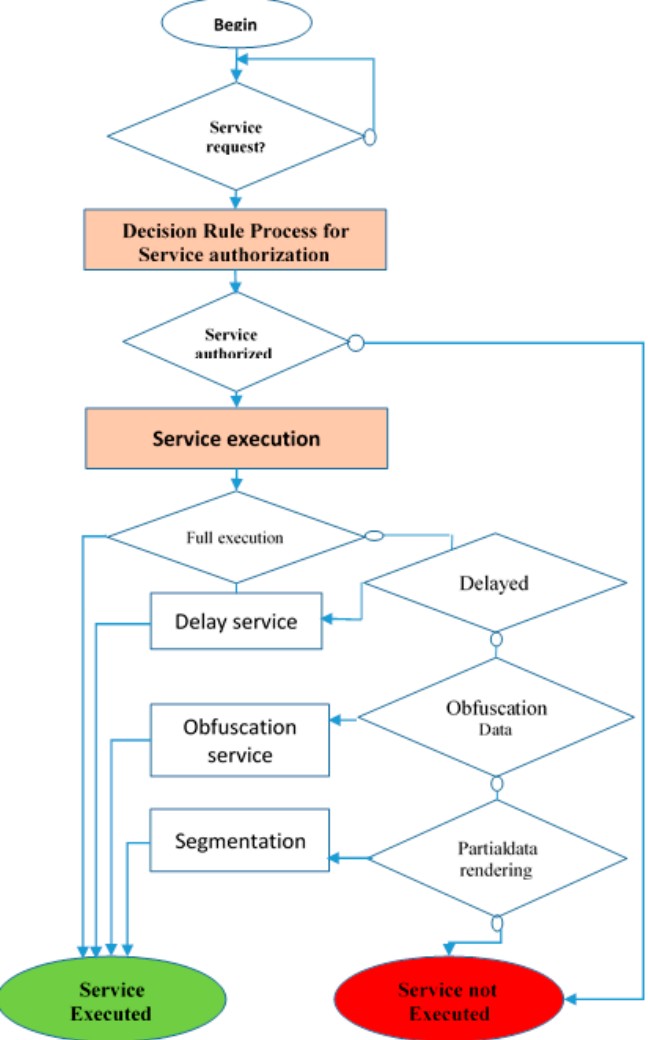

**Figure 18.** Sequence of execution of a service.

### 6.1.1. Decision Rule Process for Service Authorization

The progress of this step is done according to Algorithm 1. This algorithm includes the entry ID of the requesting object and the requested service. It gives two types of Service Authorized and Service Not Authorized results. These results can be deduced from the Matrix of influence of relations and principles.

---

**Algorithm 1:** Service Authorization

---

**Input:** Requester ID, Service requested
**Output:** Result evaluation
If ((P = Com AND (R = OOR OR R = POR)) OR (P = Re AND (R = C-LOR OR R = SOR))) Then
Service Authorized = NOK
Else Service Authorized = OK
End If

---

### 6.1.2. Service Execution Decision Process

The progress of this step is done according to Algorithm 2. This step has as input: Advantage level (Nav), Confidence level (Nc), Payment (Pay), Equality matching (EM) and authority rank (dR).

The output is the Execution strategy (with parameters: delay, obfuscation level).

---

**Algorithm 2:** Decision Process

---

**Input:** Advantage level (Nav), Confidence level (Nc), Payment (Pay), Equality matching (EM) and authority rank (dR).
**Output:** Execution strategy
**IF** (((P = Co OR P = Sha) and (R = OOR OR R = POR)) OR ((P = Co AND ((R = C-LOR OR R = SOR) AND Nav = 1) OR (P = Co AND (R = C-WOR AND Nc > seuil)) OR (P = Do AND dR > 0) OR (P = Sha AND ((R = C-LOR OR R = SOR OR R = C-WOR) AND (dC > 0)) OR (P = Mo and Pay = 1)) **THEN**
Service Execution = Full
ELSE
**IF** (P = Re AND R = OOR) **THEN**
Service Execution = Partial Data Rendering
**ELSE**
**IF** (P = Com and (R = C-LOR OR R = SOR OR R = C-WOR)) OR (P = Re AND R = C-WOR) **THEN**
Service Execution = Data Obfuscation
**ELSE**
**IF** (P = Re AND R = POR) **THEN**
Service Execution = Delayed
**ELSE**
Service Execution = Not Provided
**END IF**
**END IF**
**END IF**
**END IF**

---

Executing Algorithm 2 results in one of the following strategies: full, Partial data rendering, data obfuscation, Delayed or Not provided.

- **Delayed service Strategy:** This step includes the following elements.

  Input: Service ID.
  Process: Determination of the time delay to apply on the service execution according to a Time Delay Strategy of the application domain. The delay can be either fixed to a simple value (e.g., *n* seconds, minutes) or it can be set free up to a high limit and to be decided by the execution service engine according to the workload of the service resource.
  Output: Service Invocation (Delay: T).

- **Obfuscated service Strategy:** It has the following elements:

  Input: Service ID.
  Process: Determination of the Obfuscation level according to the obfuscation strategy in the application domain. This can be a binary strategy i.e., a full/no obfuscation, or either

a multi-level obfuscation strategy based on a parameterized obfuscation algorithm applied on the data delivered by the service execution.

Output: Service Invocation (Obfuscation level: O).

Segmented service Strategy.

This step includes the following elements:

Input: Service ID.

Process: Determination of the Segmentation level according to the segmentation strategy in the application domain. This can be a multi-level segmentation strategy based on a parameterized segmentation function on the service execution algorithm.

Output: Service Invocation (Segmentation level: S).

## 7. Netlogo Simulation Environment

We implemented and simulated our model in the NetLogo simulation environment. NetLogo [35] was designed and written by Uri Wilensky, director of the Connected Learning and Computer-Based Modelling Centre, Northwestern University. It is a programming and simulation language for a multi-agent environment. It can be used for modelling and simulating the social Internet of Things (SIoT) environment [36]. The components of the NetLogo environment are interface, information, and procedures. The elements of the NetLogo environment are patches, turtles, and links. Each object is a turtle under NetLogo, which is a mobile agent in the platform, and it can move, rotate and change color [37].

As shown in Figure 19, each object artifact has fixed characteristics (attributes ID, AID, OID, FC), a list of services, and a dynamic table (community table). The community table records the object characteristics that are situated in its Interpersonal Distance area, and that it has previously crossed, with the established relationships (SOR, CLOR, OOR), the number of services performed and the trust level. To allow objects under NetLogo to exchange messages, we have used a "messaging.nls" extension.

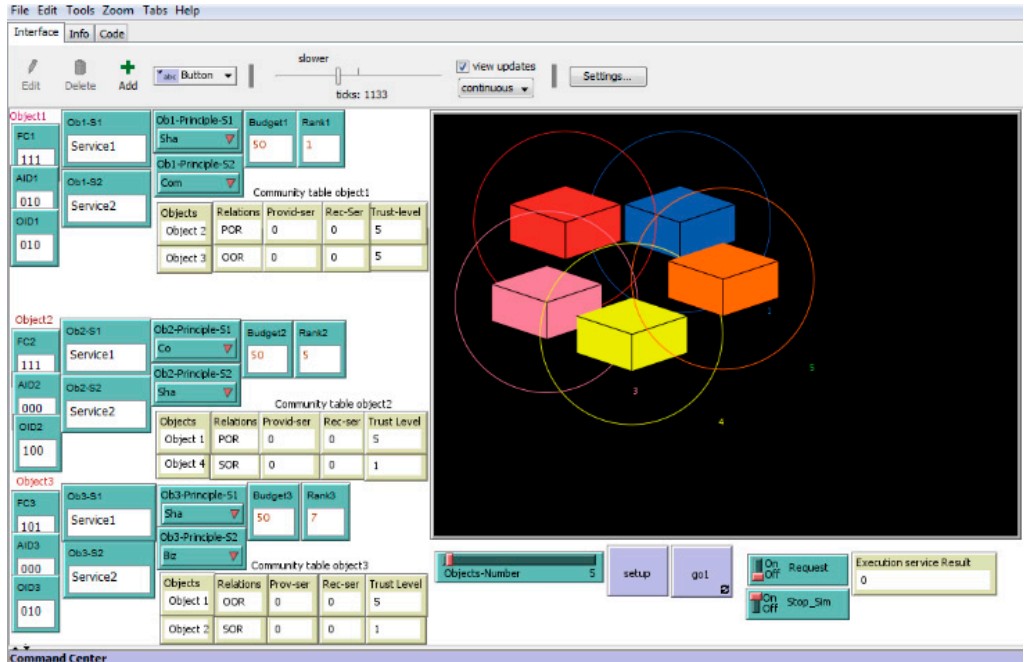

**Figure 19.** Graphical interface.

### 7.1. Creation of Objects Community

Figure 20 shows the result of a Netlogo simulation for the meeting of two objects (Object 1 and Object 2) to form a unique community. Initially, object 2 enters the interpersonal distance of object 1. On receipt of an announcement message from Object 1, Object 2 responds with a message containing a relationship proposal. This proposal is defined

after Object 2 has executed a process that determines the type of relation best suited to the characteristics of Object 1 (See Figure 5). Both objects update their community tables with the new entrance which extends the community.

```
Command Center
01:39:28.700 AM 26-nov.-2021
Simulation of 2 objects:
1009;1;--->;FF;Announcement PUSH;[111 010 010];broadcast message
1010;2;<---;1;Announcement RECV;[111 010 010];Receiving an announcement message
1011;2;--->;1;Relation Proposal SEND;[111 000 100 POR];Sends its attributes and a relationship proposal
1012;1;<---;2;Relation Proposal RECV;[111 000 100 POR];receipt of attributes and a relationship proposal
1013;1;--->;2;Relation Grant SEND;[POR];Send type of approved relationship
1014;2;<---;1;Relation Grant RECV;[POR];Reception type of approved relationship
1014;2;Community Table;[Object 1:POR 0 0 5];update of the community table
1015;1;Community Table;[Object 2:POR 0 0 5];update of the community table
1016;1;--->;2;Service Discovery SEND;[1service1 1service2];Sends Service Discovery and its list of services
1017;2;<---;1;Service Discovery RECV;[1service1 1service2];Reception of Service Discovery and its list of services
1018;2;--->;1;Service Discovery SEND;[2service1 2service2];Sends list of services
1019;1;<---;2;Service Discovery RECV;[2service1 2service2];Reception list of services
```

**Figure 20.** Execution results.

When a new object arrives in the surrounding of an existing community, the current community objects respond with a relationship proposal message. The newly arrived object responds to an agreement or a disagreement message of the type of relationship proposed. When agreed, each object then sends its service list to form and extend the common community of objects.

Two types of messages are exchanged between the objects, which are community messages and service presentation messages. Community messages comprise the announcement messages and the messages for establishing a relationship between objects (relationship type proposal message and relationship type agreement/disagreement message). The service presentation message type is used to expose the services available in an object to other objects in the community. With such a message, each object sends a description of the services it proposes to the community.

With the exchange of service presentation messages, each object in the community knows the list of available services of other community objects.

As a result of a NetLogo simulation, the number of community messages exchanged to form a single community of objects is represented by Figure 21. For example, to build a community of 6 objects, 35 community messages are exchanged, which makes it possible to update the community table owned by each object.

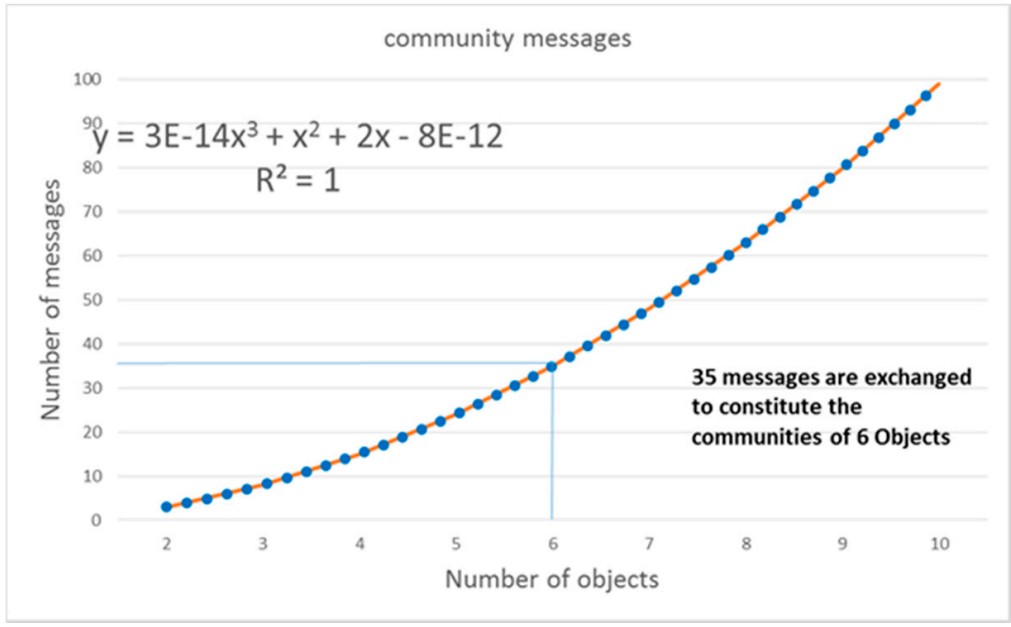

**Figure 21.** Community messages exchanged to set up a community of objects.

The linear regression estimation of the curve of Figure 21 is:

$$Y = 3 \times 10^{-14}x^3 + x^2 + 2x - 8 \times 10^{-12} \tag{7}$$

Figure 22 shows the number of service presentation messages exchanged (Axis Y) between objects regarding the number of objects (Axis X). The simulation shows that 30 messages are exchanged between all objects to create the community service lists of 6 objects.

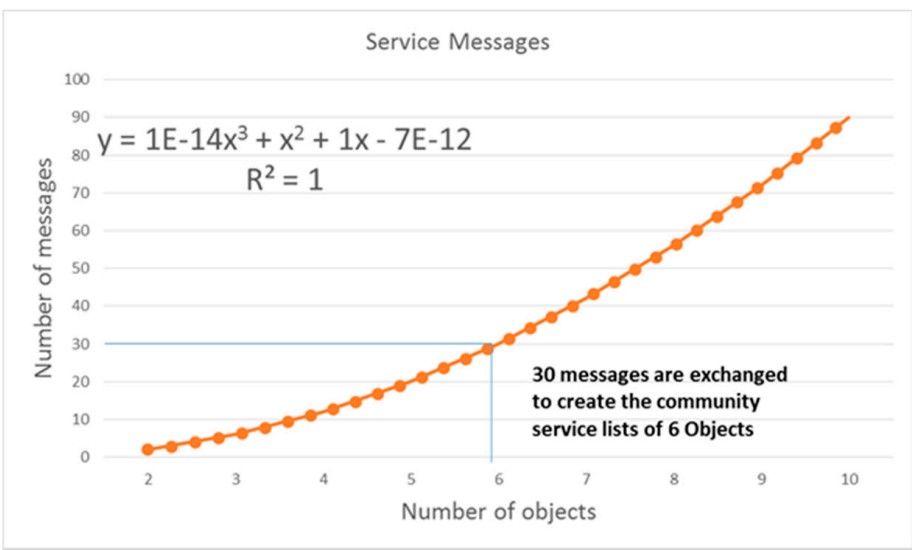

**Figure 22.** Service messages exchanged to set up a community of objects.

The linear regression estimation of the curve of Figure 22 is:

$$Y = 3 \times 10^{-14}x^3 + x^2 + x - 7 \times 10^{-12} \tag{8}$$

Figure 23 represents the total number of messages (community messages in red colour, and services presentation messages in blue colour) that are exchanged according to the number of objects in a single community.

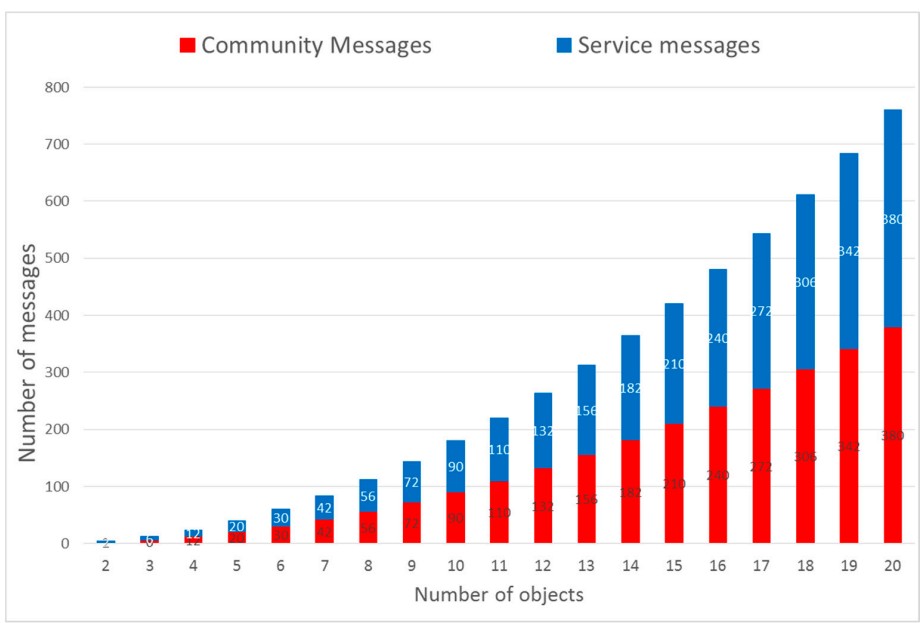

**Figure 23.** Number of messages between objects to establish community and service tables.

We can see that the number of messages exchanged between objects of a single community increases linearly with the number of objects in that community.

We can observe that the communication load is the same for both Community message exchange and services presentation messages exchange. In some exceptional cases, the service presentation messages traffic could be greater when the MTU (Maximum Transfer Unit) of the network is small or when the number of object services is great.

Figure 23 represents the communication traffic needed to establish an opportunistic and unique community of objects that share the knowledge of all their characteristics and their available services. The network traffic load is calculable and deterministic, and is easily achievable with regards to the modern communication technologies used in IoT device communication management like WiFi, Bluetooth, ZigBee, etc.

### 7.2. Generalization of Message Traffic for the Creation of a Community of Objects

The extension of the simulation process allows us to generalize the estimation of the message exchange traffic needed for the creation of a community of objects.

As a generalization, Figure 24 shows the number of community messages exchanged (Axis Y) with up to 100 objects (Axis X) to form a unique community of objects which is a satisfactory maximum size of communicating objects for an industrial application case such as manufacturing shop floor, a logistic warehousing area, etc.

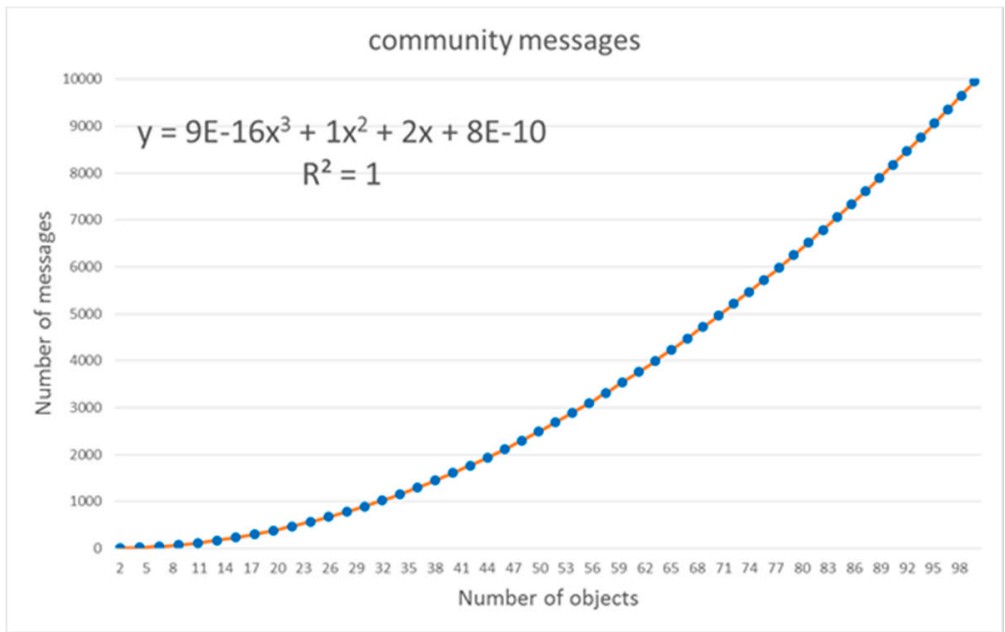

**Figure 24.** Community messages for a community of 100 objects.

Figure 25 shows the number of service presentation messages exchanged to share the list of services between all the objects in the unique community.

Thus, the number of messages necessary to create and dynamically update a community of communicating objects is deterministic. This means all messages generated by all objects to establish communities, relationships and lists of services available in the community. The number of messages is approximated satisfactorily by the formula given in Equation (9):

$$Number_{Messages} = X^2 + X \tag{9}$$

With $X$ is the number of objects in the unique community.

The number of Community creation messages ($Y$ comm) and the number of services exposition messages ($Y$ serv) are similar, as presented in Figure 26.

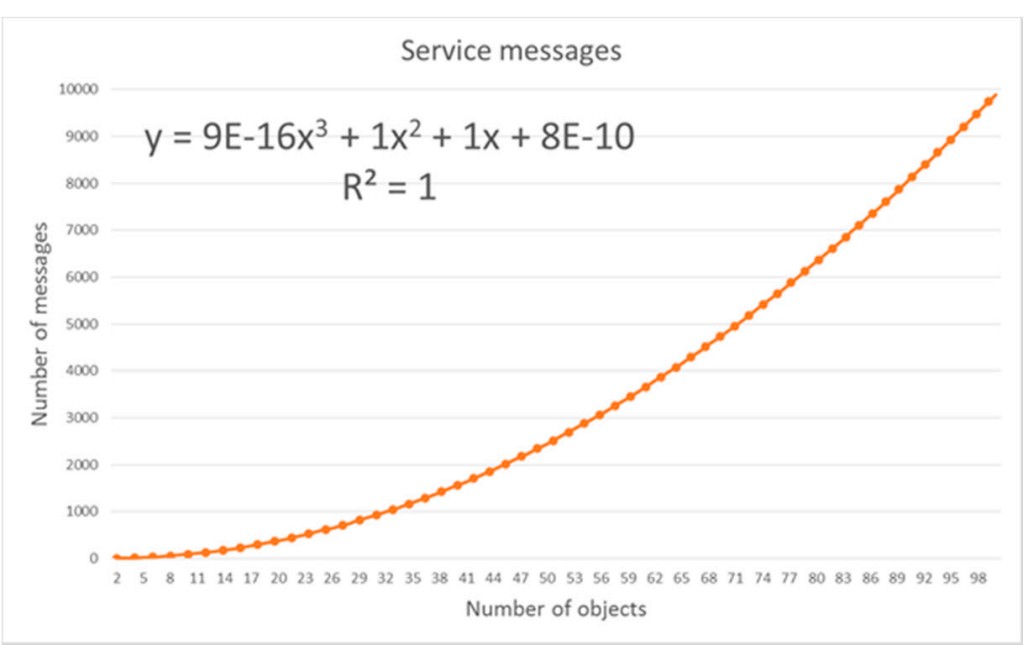

**Figure 25.** Service messages for a community of 100 objects.

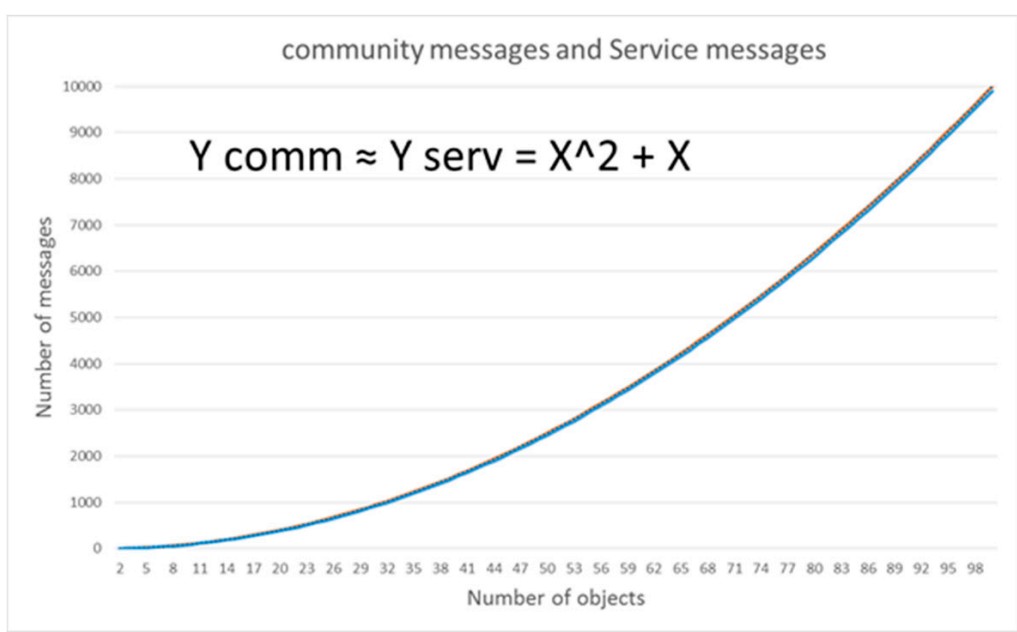

**Figure 26.** Community and service presentation messages for a community of 100 objects.

The message traffic generated during the formation of a single community of objects varies according to the number of objects in the community according to the same Equation (9) presented before. For example, to form a unique community of 56 objects, the total number of community and service presentation messages to be exchanged is 6160 to form the unique community of objects. But for the 57th object to enter an existing community of 56 objects, i.e., for it to enlarge its own community and for the other 56 objects to update their communities, it only takes 112 messages from community and 112 service presentation messages. Thus, the traffic generated between objects to form a community of socialized industrial objects is deterministic, easily computable, and it can be calculated a-priori.

The service presentation community message traffic allows the full sharing of characteristics of all objects and of their services in their communities. The complete sharing of objects characteristics and services description introduces complete and permanent

visibility of the entire community's service environment for each of its members. If the community is stable, that there are no new objects coming in, then the traffic is limited only to service requests and no longer to community messages and service presentation messages. In this case the objects interact only with service execution requests in a socialized industrial objects community which is ruled by social principles and cooperative relationships in an opportunistic way.

## 8. Conclusions

In this paper we have proposed an innovative socio-inspired methodology and associated model of interaction between industrial communicating objects. Our proposal is based on the transposition of human social interactions mechanisms and principles towards the industrial communicating objects we encounter in manufacturing and logistics facilities.

The article describes how industrial objects can interact according to an innovative and sociologic service interaction mechanism, conditioned by the imitation of social behaviours inspired by research work in sociology on interpersonal human interaction. The innovative model named CoDoComShaRe.Biz relies on six principles of interaction inspired by major social and anthropological research works on human beings' social interactions. Objects interaction is carried out by the application of the SIoT concept, i.e., the response to services is made through principles of social interaction according to strategies and conditions. The CoDoComShaRe.Biz model features relationship-based mechanisms and algorithms to drive interactions between industrial objects to achieve their individual goals. Socialized industrial communicating objects form a community in an autonomous and dynamic way by exchanging two types of messages to perfectly know each other, which are community messages and service presentation messages.

Service presentation messages and community presentation messages enable comprehensive, adaptive, and dynamic sharing of the characteristics and service description of all objects in object communities. Each object community dynamically adapts to new incoming objects.

A Netlogo simulation showed the efficiency and practicability of our model by elaborating the socialized interactions and calculating all messages generated by the objects used to establish communities, relationships and lists of services available in the community. The results demonstrate that messaging traffic between socialized industrial communicating objects is deterministic and can be estimated a priori with a simple equation. Messaging can be easily implemented and supported by current wireless sensor network communication technologies used in industrial applications.

The next research challenge is to extend the interaction model with tiny, embedded machine learning algorithms to optimize features of social interactions based on historical behavior. In addition, threats related to cybersecurity must also be investigated in future works, because autonomous communicative objects in a SIoT architecture remain sensitive elements to cyberattacks, and the resilience of the proposed model and methodology can be enhanced.

**Author Contributions:** Conceptualization, R.K., A.Z. and E.B.; methodology, R.K., A.Z. and E.B.; software, R.K., A.Z. and E.B.; validation, R.K., A.Z. and E.B.; formal analysis, R.K., A.Z. and E.B.; investigation, R.K. and E.B.; resources, R.K., E.B. and M.N.A.; data curation, R.K., A.Z. and E.B.; writing—original draft preparation, R.K., A.Z., E.B. and M.N.A.; writing—review and editing, R.K., A.Z., E.B. and M.N.A.; visualization, R.K.; supervision, R.K., A.Z., E.B. and M.N.A.; project administration, A.Z., E.B. and M.N.A.; funding acquisition, E.B. All authors have read and agreed to the published version of the manuscript.

**Funding:** This research received no external funding.

**Conflicts of Interest:** The authors declare no conflict of interest.

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
