# Peer review of "A Socio-Inspired Methodology and Model for Advanced and Opportunistic Interactions between Industrial IoT Objects"

_electronics, doi:10.3390/electronics11081281_

Round 1

Reviewer 1 Report

Broad comments. The authors have made a concise overview of the topic and a brief reference to existing literature. They have indicated the main task of the paper among its motivation. Finally, they have pointed out the key message and the potential benefits of their work.

Specific comments. In general, the text is very well structured and has clearly defined topics. Some comments for improvement:

  1. As a general remark, I could say that authors should explain all abbreviations the first time they appear, even for well-known ones.
  2. Authors could consider refining the abstract in a way that better describes the main outcome and novelty of their work.
  3. In addition, authors should make sure that the motivational impact and novelty of their work are well described in the introduction.
  4. The manuscript needs some refinement. It seems that some words have been left from the template, during the preparation phase of the work. For example, the work starts with the words “This document”, or in the second sentence (line 27) the word he may not be relevant.
  5. While authors present a brief review of IIoT challenges and environments where the distributed architecture is important, they missed some areas where this architecture is vital such that of the maritime sector [1], or Oil and Gas [2].

[1] S. C. Christos, T. Panagiotis and G. Christos, "Combined multi-layered big data and responsible AI techniques for enhanced decision support in Shipping," 2020 International Conference on Decision Aid Sciences and Application (DASA), 2020, pp. 669-673, doi: 10.1109/DASA51403.2020.9317030.

[2] S. Nikolaidis, D. Porlidas, G. -O. Glentis, A. Kalfas and C. Spandonidis, "Smart sensor system for leakage detection in pipes carrying oil products in noisy environment: The ESTHISIS Project," 2019 29th International Symposium on Power and Timing Modeling, Optimization and Simulation (PATMOS), 2019, pp. 125-126, doi: 10.1109/PATMOS.2019.8862111.

  1. Table 1 could be shortened.
  2. Do figures 1, 7, and 20 provide any added value?
  3. Figure 6 cannot be easily read.
  4. Authors should keep the same style of citation all over the manuscript. For example, in line 403 the authors use different styles.
  5. Authors should use the same language all over the work. For example Table 7 legend is in France.
  6. Authors could consider adding a reference for the Netlogo simulation environment.
  7. Authors could consider adding legends in axes in Figures 21 and 22.
  8. In addition figure 23 y-axis’ legend needs refinement. Is the title of the figure needed?
  9. Equation (9) in line 590 cannot be easily read
  10. Conclusions should be descriptive of the results but also illustrate the borders and limitations towards a fully SIIoT architecture. Threats (e.g. cyberattacks) could also be added.

Author Response

We thank you for your time on this reviewing and beneficial corrections. Please find below responses from the authors.

Best regards

Reviewer 2 Report

-the introduction must be extended, you must highlight the research problematic. Furthermore, at the end of intorduction, you should seperate your contirubtions from paper outline. list your contributions in the form of bullet points.

-the title must be changed, remove 'CoDe ShaRe.Biz' it is not adequate at all, use shorter acronym or no acronym at all. i suggest the following title: 
' a socio-inspired methodology and model for advanced and opportunistic interactions between industrial IoT objects'
and use shorter acronym in the main text, such as 'CodeSH' or 'ShaRe'

-the figure quality must be improved, some figures are not clear at all.
-in page 16, the equasions are not clear at all, eq 2, looks like it was pasted as image, additionally, in eq3 燨????? 
same thing for eq 9, please fix that, never submit paper without double checking such obvious typos.

-add the following related references:
[1] "The social internet of things (siot)–when social networks meet the internet of things: Concept, architecture and network characterization." Computer networks 56.16 (2012): 3594-3608.'
[2] "IoT-enabled social relationships meet artificial social intelligence." IEEE Internet of Things Journal 8.24 (2021): 17817-17828.
[3] "From" smart objects" to" social objects": The next evolutionary step of the internet of things." IEEE Communications Magazine 52.1 (2014): 97-105.
[4] "STLF: Spatial-temporal-logical knowledge representation and object mapping framework." 2016 IEEE International Conference on Systems, Man, and Cybernetics (SMC). IEEE, 2016.

Author Response

(The authors gave the same response as above.)
